# China's coal mine methane regulations have not curbed growing emissions

Scot M. Miller[1,5], Anna M. Michalak [1], Robert G. Detmers[2], Otto P. Hasekamp[2], Lori M. P. Bruhwiler[3] & Stefan Schwietzke [3,4,6]

Anthropogenic methane emissions from China are likely greater than in any other country in the world. The largest fraction of China's anthropogenic emissions is attributable to coal mining, but these emissions may be changing; China enacted a suite of regulations for coal mine methane (CMM) drainage and utilization that came into full effect in 2010. Here, we use methane observations from the GOSAT satellite to evaluate recent trends in total anthropogenic and natural emissions from Asia with a particular focus on China. We find that emissions from China rose by $1.1 \pm 0.4$ Tg $CH_4$ yr$^{-1}$ from 2010 to 2015, culminating in total anthropogenic and natural emissions of $61.5 \pm 2.7$ Tg $CH_4$ in 2015. The observed trend is consistent with pre-2010 trends and is largely attributable to coal mining. These results indicate that China's CMM regulations have had no discernible impact on the continued increase in Chinese methane emissions.

[1] Department of Global Ecology, Carnegie Institution for Science, 260 Panama St., Stanford, CA 94305, USA. [2] SRON, Netherlands Institute for Space Research, 3584 CA Utrecht, Netherlands. [3] Global Monitoring Division, National Oceanic and Atmospheric Administration, 325 Broadway R/GMD 1, Boulder, CO 80305, USA. [4] Cooperative Institute for Research in Environmental Sciences, University of Colorado Boulder, 216 UCB, Boulder, CO 80309, USA. [5] Present address: Department of Environmental Health and Engineering, Johns Hopkins University, 3400 N. Charles Street, Baltimore, MD 21218, USA. [6] Present address: Environmental Defense Fund, 2060 Broadway, Boulder, Colorado 80302, USA. Correspondence and requests for materials should be addressed to S.M.M. (email: smill191@jhu.edu)

China is the world's largest producer and consumer of coal (ref. [1], Supplementary Fig. 1), and coal accounts for ~72% of the country's electricity generation (as of 2015, ref. [2]). This reliance on coal has widely-recognized, adverse impacts on China's air quality. For example, coal burning contributes 40% of China's total, population-weighted PM2.5 exposure and 366,000 premature deaths as of 2013[3].

China's coal consumption also has an outsized influence on global greenhouse gas (GHG) emissions. China is the world's largest anthropogenic emitter of methane gas (CH$_4$) according to some estimates[4], and the coal sector contributes the highest fraction of the country's anthropogenic CH$_4$ emissions (~33%)[4]. CH$_4$ accumulates in coal seams during the process of coalification —when organic material is slowly converted into coal over geological time scales[5]; the majority of coal-related CH$_4$ emissions occur when the coal is mined and this trapped CH$_4$ gas is released to the atmosphere (e.g., refs. [6,7]).

Coal production has been increasing in China, at least until 2015[1,8]. Coal production increased 2.5-fold between 2000 and 2010—from 1384 to 3428 million metric tons[1]. Existing emissions inventories, however, provide divergent estimates on how this increase affected China's CH$_4$ emissions. For example, the US Environmental Protection Agency (EPA) estimates a trend of 0.33 Tg CH$_4$ yr$^{-1}$ while the Emission Database for Global Atmospheric Research inventory (EDGAR, v4.3) puts the trend at 1.5 Tg CH$_4$ yr$^{-1}$ (mean trend for 2005–2010)[4,9]. Estimates based on observations of atmospheric CH$_4$, on the other hand, are relatively consistent in reporting a trend of 1.0 to 1.2 Tg CH$_4$ yr$^{-1}$ between 2000 to 2010[10–12], with only one study estimating a trend as high as 2.0 Tg CH$_4$ during a subset of these years[12]. This annual increase is larger than total annual anthropogenic CH$_4$ emissions from countries like Greece or the Netherlands[4].

The national government, however, has set ambitious benchmarks for the utilization of CH$_4$ produced during the coal mining process (referred to as coal mine methane or CMM). China's twelfth Five Year Plan specifies that total CMM utilization should have been 8.4 billion cubic meters or 5.6 Tg of CH$_4$ by 2015. Targets for 2020 are even more ambitious; CMM recovery should be 13.2 Tg CH$_4$ (20 billion cubic meters) by that date, and a large majority of this production should be utilized, not flared or vented[13]. To reach these CMM goals, beginning in 2006, the State Council required that all coal companies drain mines of CH$_4$ prior to coal production and declared that coal mines cannot legally operate without CMM drainage systems[14].

Subsequently, the national government enacted a policy requiring that mines either utilize or flare all drained CH$_4$, a policy that became effective for all coal mines beginning in 2010[14]. These regulatory requirements have been paired with financial incentives. Mine operators receive a monetary subsidy for all utilized CMM and receive a mandatory price premium for the resulting electricity that is sold to the grid. Grid companies are further required to prioritize electricity produced from CMM[14,15]. The utilized CH$_4$ is also exempt from licensing fees and royalties[16]. These policies, however, have a notable caveat. Mine operators are exempt from flaring and utilization requirements if the drained gas has a CH$_4$ content <30%. This is because CH$_4$ at concentrations between 5–16% is explosive due to the high O$_2$ to CH$_4$ ratio and is therefore dangerous to transport or flare[17].

Existing evidence indicates that these targets, regulations, and incentives for CMM flaring and/or utilization are ambitious. CMM utilization jumped from 0.6 and 2.3 Tg CH$_4$ between 2005 and 2012 (0.9–3.5 billion cubic meters respectively)[18], but this is well below the 2015 target.

In the present study, we estimate CH$_4$ emissions across temperate and tropical Asia for 2010–2015, specifically focusing on China to explore the extent to which environmental regulations and structural changes have impacted CH$_4$ emissions from the country. We do so using 6.5 years of CH$_4$ observations from the GOSAT satellite (e.g., ref. [19]) paired with a global atmospheric model and an atmospheric inversion to estimate emissions.

## Results and Discussion

**Global trends in GOSAT observations.** Figure 1 displays the trend in GOSAT observations between September 2009 and September 2015 for aggregate 2.0° × 2.5° latitude–longitude grid boxes. The trend in Fig. 1 is relative to the trend in the National Oceanic and Atmospheric Administration (NOAA) globally averaged marine monthly mean data[20]. These trends are also independent of changes in global average hydroxyl radical mixing ratios because they are relative to a global background trend.

The trend in atmospheric CH$_4$ mixing ratios in tropical Africa, sub-tropical Asia, and temperate Asia is large relative to the global mean trend (Fig. 1). Atmospheric CH$_4$ has been increasing globally since 2007 (e.g., refs. [21,22]). This trend in sub-tropical Asia and tropical Africa is consistent with existing studies that show tropical regions have been driving these recent global CH$_4$ increases (e.g., refs. [21,22]), and variability in these tropical fluxes is not well captured in existing bottom-up models (e.g., ref. [23]).

By contrast, other regions of the globe do not exhibit such clear trends relative to the global mean. For example, the trend in many areas of the US is likely nominal relative to the global marine trend (Fig. 1). Additional analysis of US CH$_4$ trends are beyond the scope of the present study. Additionally, there is no clear pattern in Fig. 1 across the Amazon or adjacent regions. Large fires in the Amazon in 2010 emitted a pulse of CH$_4$ to the atmosphere, making it difficult to fit a simple multi-year trend for the region (e.g., ref. [24]). Further, note that trends at high latitudes are highly uncertain due to data sparsity (e.g., ref. [25]), and we do not discuss these regions in detail here.

**Trends in emissions from China and Asia.** Changes in atmospheric CH$_4$ mixing ratios over a particular region can be due to

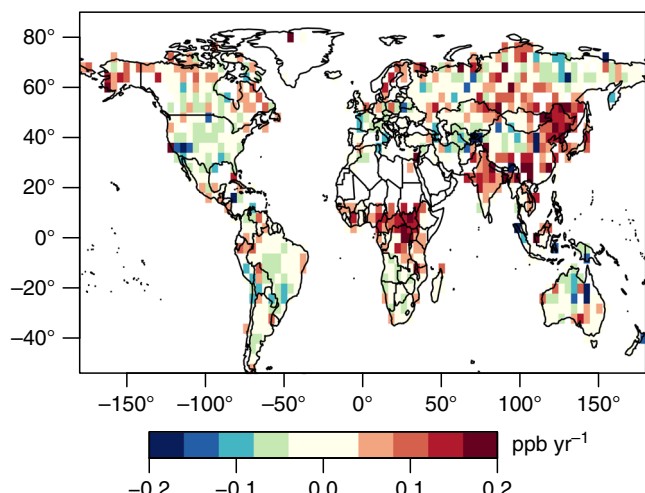

**Fig. 1** Trend in nadir GOSAT observations. The figure displays the trend between September 2009 and September 2015 minus the trend in the NOAA global marine observations. The GOSAT observations are averaged into 2.0° × 2.5° latitude–longitude boxes before fitting the trend, and the figure only displays boxes with more than 250 total observations. Red colors indicate that the GOSAT observations are increasing faster than the NOAA global marine average while green and blue colors indicate an increase slower than the NOAA average. China, India, tropical Africa, and tropical Asia show increases that are faster than the global average

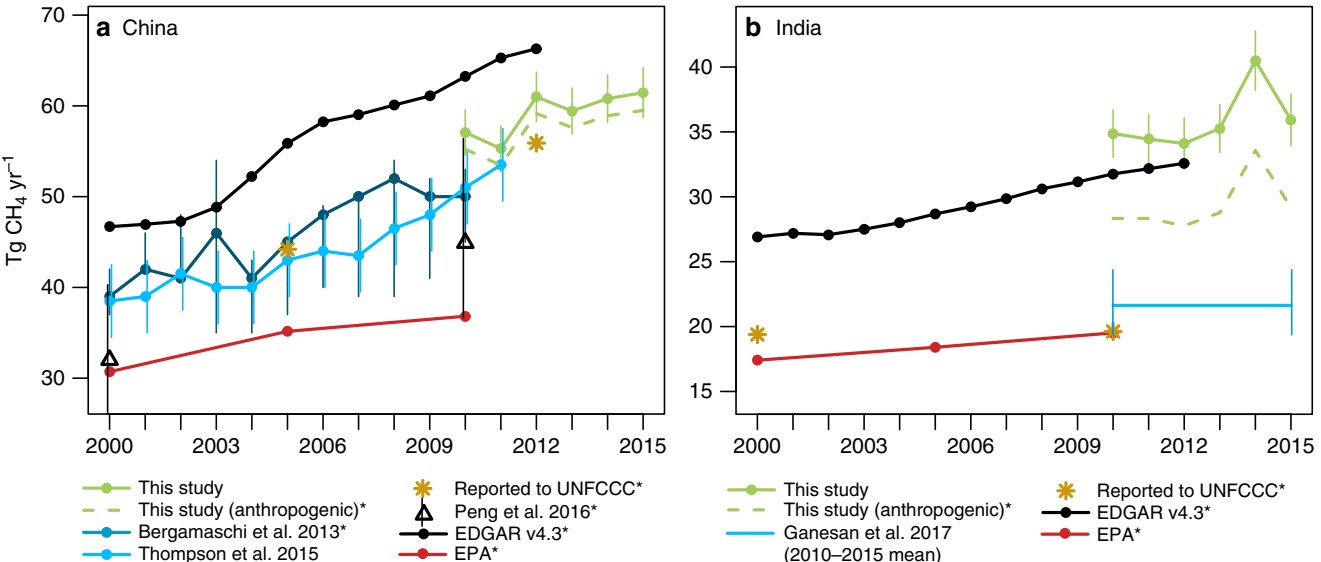

**Fig. 2** Methane emissions estimates for China and India. **a** $CH_4$ emissions estimates for China from this study, Bergamaschi et al.[10], Thompson et al.[12], UNFCCC[35], Peng et al.[6], the EDGAR v4.3 inventory[4], and the US EPA[9]; and **b** for India from this study, Ganesan et al. [28], UNFCCC[53], and US EPA[9]. We find a trend in emissions from both countries for 2010–2015, though the trend for India is uncertain. Note that uncertainty estimates for this study are 95% confidence intervals, and uncertainty bounds for Bergamaschi et al.[10] reflect the range of different inversions that use different datasets (e.g., in situ, satellite). Estimates marked with an asterisk are for anthropogenic emissions only. Furthermore, the dashed green line represents the posterior emissions estimate after subtracting the wetland emissions model, biomass burning inventory (GFED), and termite emissions

$CH_4$ emissions in that or any upwind regions, and an inverse modeling framework can be used to attribute patterns in atmospheric $CH_4$ to patterns in surface emissions.

We incorporate atmospheric $CH_4$ observations into an inverse model and find an increasing trend in $CH_4$ emissions across much of Asia, including in China and India (Figs. 2 and 3, Supplementary Figs. 2 and 3). The trend in total emissions from China is $1.1 \pm 0.4$ Tg $CH_4$ yr$^{-1}$ ($p = 0.058$). Globally, $CH_4$ emissions have been increasing at a rate of ~5 to 8 Tg $CH_4$ yr$^{-1}$ since 2007 (e.g., refs. [10,20]). The emissions increase from China accounts for ~11–24% of this total global trend (95% confidence interval). The estimated trend in Indian emissions is less certain, on the other hand, at $0.7 \pm 0.5$ Tg $CH_4$ yr$^{-1}$ ($p = 0.25$).

These results show that the reported trend in China's $CH_4$ emissions prior to 2010 has continued in subsequent years, in spite of regulations aimed at substantially reducing coal mining emissions. Top–down atmospheric studies generally indicate an annual trend of 1.0–1.2 Tg $CH_4$ for the 2000s (e.g., refs. [10–12]), and we find that a trend of the same magnitude has continued past 2010. The results of this and earlier studies collectively indicate that China's annual $CH_4$ emissions increased by ~50% between 2000 and 2015 (~20 Tg $CH_4$), an increase comparable to total annual anthropogenic $CH_4$ emissions from countries like Russia and Brazil[4]. By comparison, this increase is more modest than reported in earlier versions of EDGAR (3.3 Tg $CH_4$ yr$^{-1}$ mean trend in EDGAR v4.2 for 2000–2010) but is similar to the newer version of EDGAR (1.6 Tg $CH_4$ yr$^{-1}$ average trend in EDGAR v4.3 for 2000–2012).

The estimated emissions for India and China are also in good agreement with available in situ $CH_4$ observations across Asia. Modeled total column $CH_4$ using the estimated emissions have a smaller bias and correlate better with in situ observations relative to the prior emissions. Figure 4 displays model–data comparisons at four in situ sites—in China, Korea, and Japan. Two of these sites are mountain-top sites (panels a and c) while two are marine sites (panels b and d). None of these sites are included in the inverse model, providing an independent check on the emissions estimated using GOSAT observations.

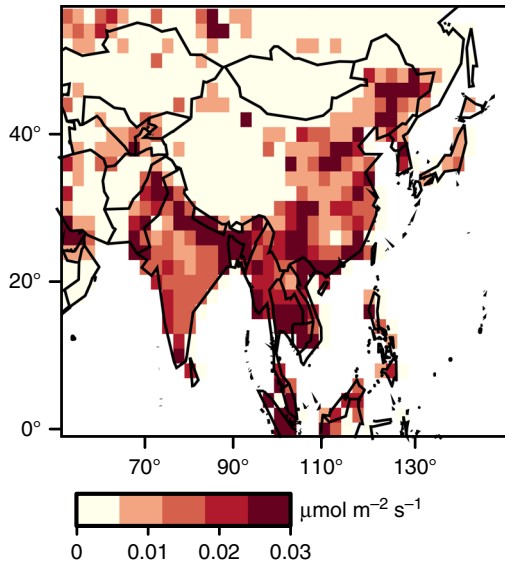

**Fig. 3** Map of $CH_4$ emissions estimates. Total $CH_4$ emissions (anthropogenic plus natural) estimated using GOSAT observations and the inverse model (2010–2015 mean). $CH_4$ emissions from China are highest in provinces with large coal production and coal formations that contain high amounts of $CH_4$ (e.g., Shanxi, Guizhou, and Anhui; refer to Supplementary Fig. 1). Note that the inverse modeling emissions estimate is highly uncertain for any individual grid box, but those uncertainties decrease at increasing spatial scales (Supplementary Fig. 2)

Note that the total emissions estimate for India is in the mid-range of existing, global inverse modeling studies[26,27]. These earlier studies compare results from multiple global inverse models and report multi-model averages of 33 and 39 Tg yr$^{-1}$, respectively (for 2000–2009 and 2003–2012, respectively). The emissions reported here ($36 \pm 2.5$ Tg yr$^{-1}$) are consistent with those studies. By contrast, a recent, regional inverse modeling

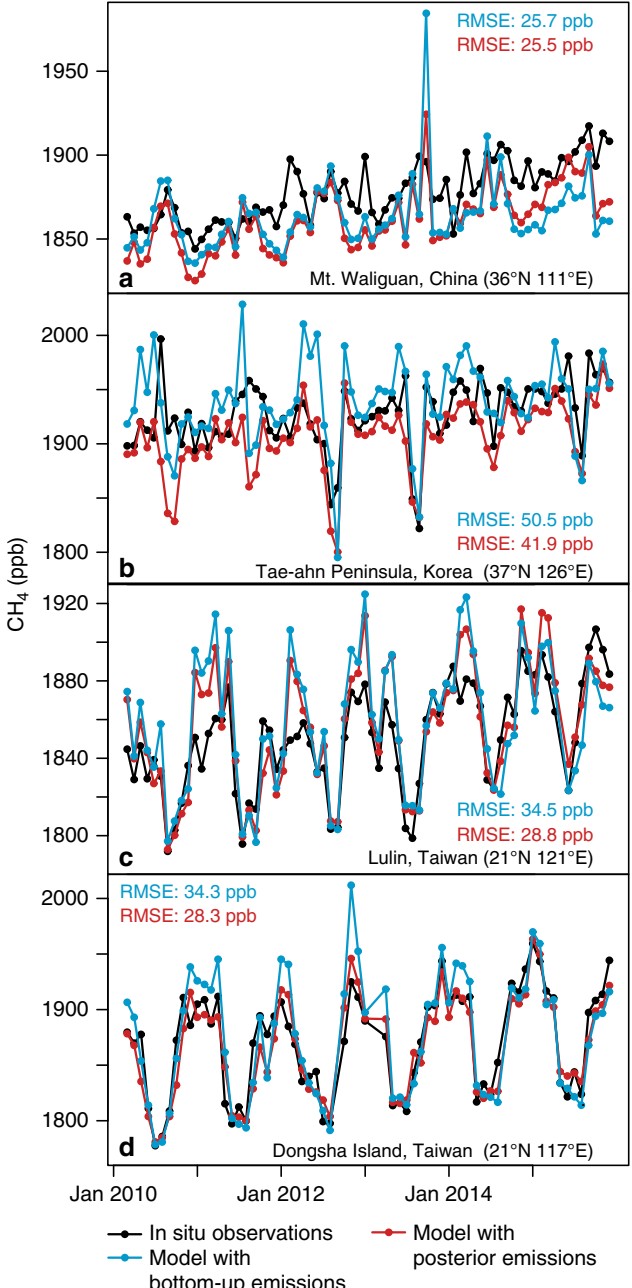

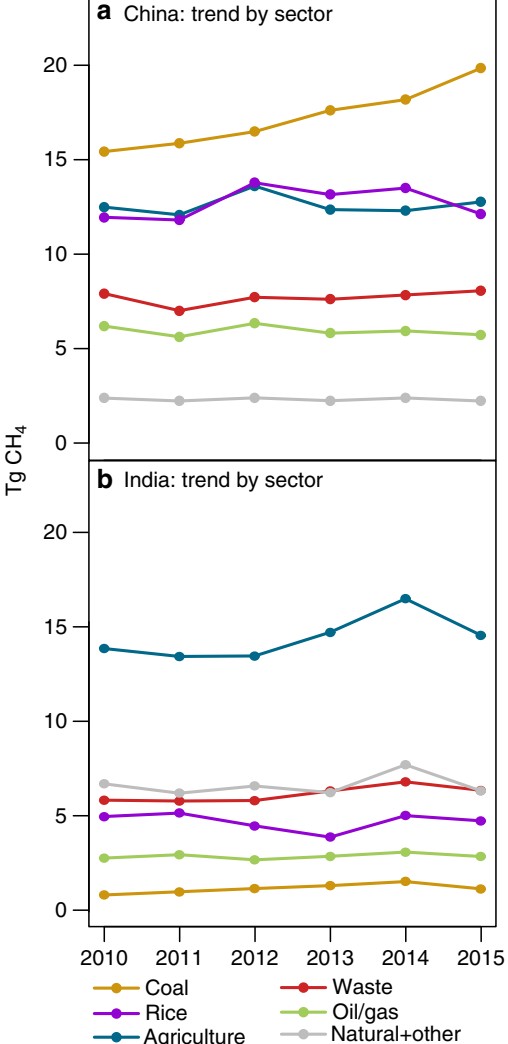

**Fig. 4** Observed and modeled $CH_4$ in situ mixing ratios. The sites shown are in situ monitoring stations from the NOAA Global Greenhouse Gas Reference Network[20]. Modeled $CH_4$ using the posterior emissions shows a lower root mean squared error (RMSE) at all sites. Note that all panels display monthly model and data means, and the blue modeled time series in this plot use EDGAR v4.2, the same inventory version used in the inverse model

**Fig. 5** Emissions trends by sector. **a**, **b** show the estimated emissions trend by sector for China and India, respectively. The coal sector appears to be driving the trend in China. No clear trends are obvious for India. Note that all trends in this figure are driven by the GOSAT observations, not by the EDGAR anthropogenic emissions inventory used in the inverse model; the inventory estimate is constant with time. The EDGAR inventory does not include uncertainty estimates for the sector-specific breakdown of emissions, and uncertainty estimates are therefore not included here. Figure 3 and Supplementary Fig. 2 present uncertainties for the total emissions

study of India is an outlier compared to these studies at 22 Tg yr$^{-1}$ (Fig. 2, 2010–2015 mean)[28].

**Contribution of various source sectors**. We find a clear trend in $CH_4$ emissions from China's coal sector while other source sectors do not show a corresponding trend (Fig. 5). By contrast, no source sector in India shows an obvious trend; the trend in total emissions from India is uncertain, and it is not clear what could be driving that trend, if one exists. This attribution is based upon

the emissions estimate from the inverse model and the spatial distribution of different source sectors within the EDGAR emissions inventory. Specifically, we attribute emissions within each individual model grid box based upon the relative fraction of emissions that are due to each sector within that grid box in the EDGAR emissions inventory. Refer to the Methods for additional detail.

Additional lines of evidence also indicate that coal is likely driving the overall trend in China's emissions. Coal production in China increased between 2010 and 2015 (from 3400 to 4000 million metric tons[1]) whereas ruminant populations and rice production have remained flat or grown only slightly. For example, milled rice production grew from 137,000 thousand metric tons in 2010/2011 to 140,850 thousand metric tons in 2016/2017[29]. Beef production increased by only 8% between 2011 and 2016, and China's dairy cattle inventory declined due to both

decreasing dairy demand and increasing dairy and beef imports[30,31].

**Implications for coal mine methane.** Overall, results indicate that $CH_4$ emissions from China have been increasing since 2010 and that this increase has not slowed. Existing bottom–up and top–down studies disagree on the magnitude and trend in $CH_4$ emissions from China (e.g., refs. [26,27]), and the present study sheds additional light on these emissions. We find that, although China has set ambitious benchmarks, regulations, and incentives for CMM drainage and utilization since the mid-2000s, emissions continue to increase following a business-as-usual scenario. This increase in emissions is most likely driven by the coal sector, implying that China's ambitious coal $CH_4$ actions have not produced a detectable change in the rate of increase in $CH_4$ emissions.

Existing studies from the US EPA and the International Energy Agency (IEA) have identified three broad barriers that China would need to overcome to meet its CMM targets (e.g., refs. [14,15,32]). One or more of these barriers has presumably hampered China's progress, and these studies help place the results presented here within a broader policy context.

First, insufficient infrastructure makes it difficult to bring CMM to market, and the US EPA cites this challenge as a potential barrier to achieving China's CMM goals[15,32]. Most coal mines are located in remote mountainous areas, areas that are poorly connected to cities or natural gas infrastructure (ref. [32], ch. 7). Furthermore, the US EPA describes China's gas market as "underdeveloped", and only 22% of China's non-rural population had access to natural gas as of 2010[15].

Second, inadequate technology likely presents an obstacle. Most coal mines in China are deep, and the coal seams are highly impermeable, unlike many mines in the US and Australia. The CMM drainage technology often used in China is poorly suited for these conditions[14,32]. As a result, the resulting CMM is often of poor quality (i.e., low $CH_4$ content), according to US EPA[15]. In addition, the IEA explains that operators of small and medium mines often lack the technical expertize to utilize the $CH_4$ for heating or electricity production[14].

Third, inadequate or poorly-designed policies may stand in the way of reaching CMM utilization targets. US EPA explains that existing regulations and incentives have not been fully realized, and some may have backfired[15]. Utility companies often resist accepting electricity generated from CMM, in spite of policies that require utilities to give priority to this electricity. According to US EPA[15], "The incentive program for CMM power plant utilization proved particularly difficult to implement due to resistance from power grid companies uneager to manage the complexities of dispatch of the fluctuating output of small CMM plants, and lacking a policy mechanism to pass the premiums through to consumers." CMM utilization requirements may have also backfired. Government policy requires that all mines utilize drained gas with greater than 30% $CH_4$ content. EPA has anecdotal evidence that mine operators may be diluting drained gas to circumvent the requirement[15]. These actions not only render CMM unusable but also unsafe. In addition, the IEA points out that most local and provincial governments have limited power to enforce CMM regulations, limiting overall enforcement action[14].

Existing inventories diverge on how China's coal $CH_4$ emissions have changed since 2010 (e.g., refs. [4,9]). Emissions factors (i.e., leak rates) provide a convenient means to compare these inventory emissions and estimated trends. Emissions factors in existing inventories range from ~5 to 11 $m^3$ of $CH_4$ per metric ton of coal mined (weighted national average) (Fig. 6). We find

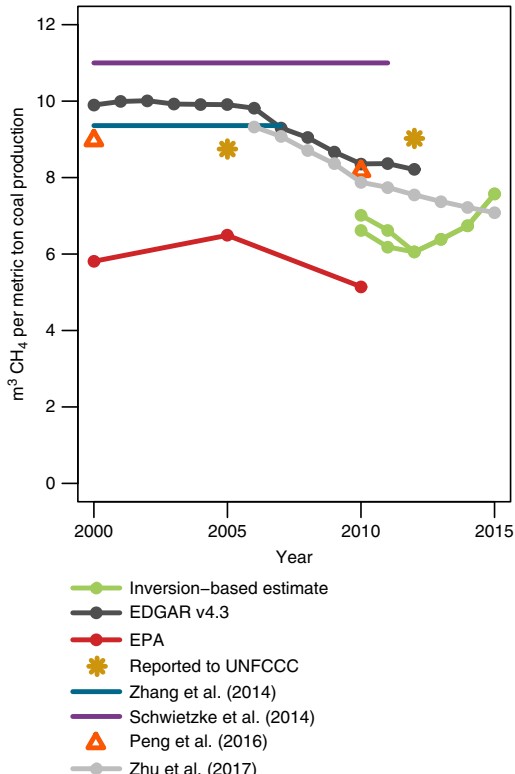

**Fig. 6** Coal $CH_4$ emissions factors. Emissions factors implied by this study, EDGAR v4.3[4], US EPA[9], UNFCCC[35], Zhang et al.[54,] Schwietzke et al.[55,] Peng et al.[6,] and Zhu et al.[7]. The factors implied by this study are in the lower mid-range of existing estimates. In some cases, we divide total emissions by coal production numbers to derive emissions factors for this figure. For example, we divide the EDGAR emissions estimate by EIA coal production estimates[1], and we divide Peng et al.[6] and China's UNFCCC report by production numbers from China's Statistical Yearbook[8], the production numbers used in those estimates. The two green lines in this figure are the inversion estimate divided by production numbers from EIA and China's Statistical Yearbook, respectively. Note that Schwietzke et al.[55] and Zhang et al.[54] use time-invariant emissions factors, and these studies are therefore represented with solid lines

emissions factors of ~6–7 $m^3$ of $CH_4$ per metric ton, depending upon the year, by dividing coal emissions estimates presented here (Fig. 5) by China's total coal production[1,8]. Note that we divide China's official emissions estimate by its own coal production numbers while we divide the EDGAR emissions inventory by coal production numbers from the Energy Information Administration (EIA)[1,8]. These two coal production estimates contain similar trends but differ by up to ~5–10% in some years.

These emissions factors are similar to those in several existing inventories, but the trend is not. The emissions factors in many inventories have been declining with time and are forecasted to continue declining in future years (Fig. 6). These declines are often due to assumptions about improved energy technology and the forecasted effects of environmental regulations (e.g., ref. [7]). By contrast, the emissions factors implied by this study show a slight upward trend from 2011 onward. This upward trend could be real, it could be due to uncertainties or errors in the inverse model and the associated source attribution, or it could point to inaccuracies in China's coal production statistics. The EIA's coal production numbers for China are uncertain, and there have recently been large discrepancies and an upward revision in China's coal production and consumption statistics[33,34]. If the

EIA were to underestimate China's production trend, it could alias into the emissions factors and create a spurious upward trend. Note that the emissions factors from China's National Bureau of Statistics also show a slight upward trend between 2005 and 2012. China submitted an explanation of its provincial coal emissions factors in 2012 along with its emissions estimate for 2005[35]. We suspect that China used the same emissions factors in its 2012 emissions update, and changes in the emissions factor between 2005 and 2012 in Fig. 6 more likely reflect a shift in coal production to provinces with higher coal $CH_4$ content.

Overall, we find that China's $CH_4$ emissions have continued to increase unabated since 2010, likely driven by increasing coal production (Fig. 5). Specifically, estimated $CH_4$ emissions increases are highest in regions where the EDGAR inventory indicates a predominance of coal mining emissions relative to other source types (Supplementary Figs. 1 and 3, refer to Methods). Furthermore, coal production in China continues to increase while cattle counts and rice production have remained relatively flat during the study period. These results imply that China's regulations and initiatives have not produced a detectable flattening or decline in $CH_4$ emissions. China has an opportunity to mitigate substantial $CH_4$ emissions through CMM drainage and utilization (or flaring), and the national government in China has taken steps to require more environmentally friendly practices. Observations from GOSAT indicate a business-as-usual emissions scenario up to 2015, and it is therefore unlikely that China has met its ambitious regulatory goals.

## Methods

**GOSAT observations**. The Greenhouse Gases Observing Satellite (GOSAT) was launched in January of 2009 by the Japan Aerospace Exploration Agency (JAXA). The satellite flies in a sun-synchronous polar orbit, passing each location at ~13:00 local time (e.g., ref. [19]). GOSAT is sensitive to $CH_4$ mixing ratios throughout the troposphere, and this sensitivity slowly declines in the stratosphere at increasing altitudes (e.g., ref. [36]).

All of the analyses conducted in this paper use $CH_4$ observations generated using the RemoTeC v2.3.8 proxy retrieval (e.g., ref. [37]). The v2.3.8 retrieval and its evaluation are described in detail in Hasekamp et al.[38] and Supplementary Note 1. A study comparing this retrieval against TCCON observations[39] indicates that the two types of observation are in good agreement[37]. Furthermore, inverse modeling estimates based on data from this retrieval are consistent with estimates based on data from the global network of in situ $CH_4$ observations[40]. We use high gain nadir observations and exclude glint observations along with any observations that have a negative quality control flag. The resulting dataset has an average of $2.2 \times 10^5$ observations per year during the study period (e.g., Supplementary Fig. 4).

The GOSAT observations and modeled total column mixing ratios show a latitude-dependent difference or bias (Supplementary Note 2). Two previous modeling studies found a comparable bias[41,42]. This bias could be due to the GOSAT observations or the atmospheric model. Existing studies have not pinpointed a cause but speculate that the bias may be due to model–GOSAT differences in the stratosphere. We apply a latitude-dependent correction to the GOSAT observations to remove this bias. We use the same procedure as a recent study[42] and describe the correction in greater detail in Supplementary Note 2 and in Supplementary Figs. 5 and 6. This correction ranges from approximately −10 to −15 ppb across East Asia, depending upon the latitude.

**Inverse model**. We use the GEOS-Chem (Goddard Earth Observing System—Chemistry) model to simulate atmospheric transport as part of the solution of the inverse problem (e.g., refs. [42,43]). Supplementary Note 3 describes these simulations in greater detail.

We use a combination of inventory estimates within the inverse model. The EDGAR emissions inventory (Emission Database for Global Atmospheric Research, version 4.2) serves as the anthropogenic emissions estimate[44], an online wetland model provides daily wetland emissions[43], and daily biomass burning emissions are from the Global Fire Emissions Database version 4[45,46]. The anthropogenic inventory used in the inversion (EDGAR)[4] is time-invariant for the setup here, such that any estimated emissions trends are solely due to the GOSAT observations.

The inverse model will scale the total emissions (anthropogenic plus natural) in each model grid box (2.0° × 2.5° latitude-longitude) within Asia and will estimate a different emissions scaling factor for each TransCom region (e.g., ref. [47]). This setup provides computational savings outside the region of interest, and the global scope of the inversion also ensures that the air masses entering the domain of

interest are consistent with global observations. We estimate a different set of scaling factors for each 6-month block of the 6.5 year study time period.

The inverse model employed here is Bayesian. It accounts for errors in the model and measurements (i.e., model-data mismatch errors) and errors in the prior emissions estimate. We also account for spatial and temporal covariances in both the model-data mismatch and prior errors by including off-diagonal elements in the associated error covariance matrices. This setup ensures more realistic uncertainties on the estimated fluxes. Supplementary Notes 4–7 describe this setup and the uncertainty calculations in greater detail. In addition, Supplementary Note 8 discusses the inverse model in context of $CH_4$ isotopes.

Supplementary Note 9 provides further discussion of how global changes in the hydroxyl radical (OH) could impact the inverse modeling results. Specifically, it is unlikely that the emissions trends estimated in this study are due to a trend in OH. Some studies argue that OH may be changing[48,49] while other studies find no evidence for recent changes in OH[50,51]. Even if OH levels were changing, the resulting trend in atmospheric total column $CH_4$ would likely be small relative to the regional trends across China in Fig. 1.

**Sector attribution**. We also investigate which emissions sectors are most likely to be responsible for emissions from China and India, and more importantly, which sectors may be driving any emissions trends (Supplementary Note 10). Within each grid box, we attribute the same fraction of emissions to each source type as in the corresponding grid box of EDGAR (e.g., refs. [42,52]). This approach does not assume that EDGAR estimates correct emissions totals for each grid box. However, it assumes that EDGAR attributes the correct fraction of emissions by sector within each grid box. Two different errors could bias this fraction and create uncertainty in the source attribution in this study: inaccuracies in the spatial distribution of each source sector and an over or underestimation of an individual source sector. The former problem (incorrect spatial distribution) appears unlikely; the spatial distribution of coal mining in EDGAR is likely accurate for China because the locations and characteristics of coal mining regions in China are well-known. We investigate the latter issue (over- or underestimate a source type) in detail in Supplementary Note 10 and Supplementary Fig. 7. We conclude that this issue does add some uncertainty to the source attribution, but the overall conclusions are robust.

## Data availability

All of the satellite and in situ $CH_4$ data used in this paper are publicly available online. The RemoTeC $CH_4$ proxy retrievals (v2.3.8) are available online at ftp://ftp.sron.nl/pub/pub/RemoTeC/C3S/CH4_GOS_SRPR/V2.3.8/. Furthermore, in situ data are archived on the NOAA Global Greenhouse Gas Reference Network: https://www.esrl.noaa.gov/gmd/ccgg/ggrn.php. Other data products used in the study like the EDGAR emissions inventory and Global Fire Emissions Database are available for download at http://edgar.jrc.ec.europa.eu/ and https://www.globalfiredata.org/.

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

## Acknowledgements

We thank Ed Dlugokencky of the US National Oceanic and Atmospheric Administration, Maria Mastalerz of Indiana University, and Clark Talkington of Advanced Resources International for their advice on the project and manuscript. We thank Lingxi Zhou for in situ observations from Mt. Waliguan, China. Rob Detmers acknowledges funding from the ESA Climate Change Initiative Greenhouse Gases project. This work is funded by the Carnegie Distinguished Post-doctoral Fellowship and NASA grant #NNX13AC48G.

## Author contributions

S.M.M. and A.M.M. designed and conducted research; R.G.D. and OP.H. developed satellite retrievals; L.M.P.B. and S.S. helped analyze results; all contributed to writing the paper.

## Additional information

**Competing interests:** The authors declare no competing interests.

