## [Peer Review File · Nature Communications]

Reviewers' comments:

Reviewer #1 (Remarks to the Author):

Comments:

The topic fits the scope of the journal, and research work presented in this paper may have significant implications. The developed dataset could also be valuable for others working on related topics.

This manuscript however do have a few places that may need to be improved:

- a. This manuscript is reasonably well-written. Though there are still room for further writing improvements.
- b. The introduction, whilst informative, is very long and should be reduced and prioritized.
- c. Where did you refer 72% in line 17, page 1.
- d. There are lots of contents in the method section. To better guide readers, I would recommend rewriting/adding one paragraph at the beginning of section 2, 3 and 4 to provide an overview of the method subsection, and to help readers organizing their thoughts.
- e. Please define the units here where you first mention them, as opposed to further down the page.
- f. References should be listed in the order in which they are quoted. Where are the reference 18th and 19th?
- g. In supplemental information, Page 1, the last line, please delete "both". Where are Fig.2b and panel (b)?
- h. What are these errors? Are these just proportional differences? Please define
- i. Please add a discussion on the limitations of the study, for completeness. In particular the data is very opaque, so a discussion around this would be useful.
- j. In line 110, "Additional lines of evidence also indicate that coal is likely driving the overall trend in China's emissions (from 3,400 to 4,000 million metric tons)." Actually, the data on coal production should be questionable.
- k. Many readers may aware that China has been aggressively targeting small scale coal mining operations and have closed many small coal mines. The trend will likely to persist in the next few years. Will this policy impact the author's results?

Reviewer #2 (Remarks to the Author):

General comments

This manuscript presents an analysis of recent, 2010 to 2015, trends in methane emissions from China based on GOSAT satellite retrievals of methane. The authors find a significant increasing trend over the study period, most likely from coal, which is in accordance with “business as usual” scenarios and indicates that policy changes to reduce emissions have had no notable impact on emissions. This study is scientifically sound and generally well presented. However, there are a few minor points that need clarification before publication.

Specific comments

L30: Please state the version of the EDGAR inventory, as the estimate has been revised with each version.

L72: The trend is actually in sub-tropical Asia and temperate Asia and in tropical Africa.

L98-99: I suggest including the comparisons with observations in the manuscript, as this is an essential part of the method validation.

L104-109: I find it a bit misleading to say that this study finds a clear trend in the emission from coal since this finding is based on the total trend and the fact that in the EDGAR inventory coal is the largest source sector. So even if the share of the total emissions across all sectors is constant in time, coal would show the largest trend; what the authors find is a trend in the total which is likely owing to coal since this likely is the largest source sector.

L141-143 and Fig. 5: Interestingly the UNFCCC emission factor estimate shows an increase between 2005 and 2012, which is in contrast with the EPA and EDGAR inventories and with what is expected based on policy changes. Can the authors comment on why this may be?

Fig.1: The absence of, and in some areas even decreasing, trend in the US is important as it contrasts with several recent studies. Currently this is only discussed in the SI. Even though the focus of this study is Asia, the authors should consider moving this to the manuscript as it’s an important result.

SI, section 2: How were the OH data used in GEOS-Chem obtained, and were these pre-optimized using another tracer?

SI, section 1.3: The authors state that GOSAT displays a latitude dependent bias compared to modelled XCH₄ (using bottom-up emission estimates). Which model is this and how can the authors be sure that the model-observation discrepancy is not due to the model, such as latitudinal dependent biases in the emission inventories, in the transport or in the atmospheric loss of CH₄? How do the bias corrections change the comparison to TCCON data?

SI, section 3.1: λ , i.e. the optimized scaling factors, are the unknown quantity but μ , the prior scaling factors, must be assigned for the inversions, i.e. the prior must be known.

SI, section 6: Figure 8c&d shows a significant decrease in CH₄ mixing ratio between 2012-2014. Is the decrease reflecting global variations? What do the authors think is causing the decrease as it contrasts with the generally increasing CH₄ emissions from China.

Technical comments

L2: I don’t think “plurality” is the right term here but rather “majority”

L4: “went” should be replaced by “came”

L85: “Indian” (not India) emissions

Reviewer #3 (Remarks to the Author):

General comments

This is a very interesting and potentially controversial paper, that attempts to show that China’s efforts to curb methane emissions have not worked.

The problem is very important, with major implications for the UN FCCC Paris Agreement. Thus the paper makes an extremely valuable contribution and strongly deserves to be published.

That said, there are major weaknesses in the paper that need to be fixed or at least duct-taped with weasel words.

1. There is no mention of the isotopic evidence for a global shift to lighter values, even though Schwietzke is an author and Turner et al and Rigby et al are mentioned (and the likelihood of an OH collapse dismissed!). Increased isotopically heavy Chinese coal methane would drive $\delta^{13}\text{C}$ strongly more heavy: it is possible to accommodate this in a global budget where the fossil fuel share is dropping, but the impact on the global isotopic balance needs to be discussed.
2. There is no discussion of what is going wrong in China – as the paragraph from line 37-44 makes clear, the Chinese government has made major changes to the policy framework; Why is this policy failing?

Conclusion – yes, this is an important paper and definitely should be published. But it needs to be armour-plated against the charge that it jumps to conclusions. Thus "Accept subject to moderate-scale revision."

Specific comments

Abstract Line 2 – "plurality": maybe this word will be tough for a non-Anglophone to understand – US/UK election meaning is the largest share but not a majority, but many readers, especially from PR democracies will assume it means 'majority'. Also say anthropogenic. I know this is the quasi abstract but this statement needs to cite a reference either here or in next para – Peng et al? Atmos. Chem. Phys., 16, 14545-14562. The Peng paper should be cited somewhere: they get 45Tg anthropogenic CH₄ in 2010.

L17 – 72% in which year? Needs a ref.

L21 – world's largest emitter. Well, maybe but how about Brazil and its wetlands? In general this paper makes no distinction between natural and anthropogenic methane. That's a mistake – the paper needs to be very clear exactly when it is talking about anthropogenic emissions only, and when it means total emissions. Also, the ref here is EDGAR, but there is a huge discrepancy between EDGAR and top-down estimates.

L24-25 majority in coal mining? Is that proven? – not in power station pulverising? REF needed.

L25 "and so have coal-related emissions". – no documentation for this bald statement. It is not necessarily true – the task of the paper is to prove that China's coal industry reforms have not made it better at capturing methane leaks. This line is tantamount to saying the accused is guilty before hearing the evidence.

L35 – note the Thompson study uses isotopes. That's the big omission in this paper.

L63 and L70-75 – GOSAT - worth mentioning Parker, R.J., et al. (2018) Evaluating year-to-year anomalies in tropical wetland methane emissions using satellite CH₄ observations. Remote Sensing of the Environment, 211, 261-273.

Fig. 1 – probably should have a link to Supplementary online material about the methodology of GOSAT retrievals, or at least a 'go-to' reference. L

Fig. 2 – difference with the Ganesan study is very interesting. Needs some discussion in text somewhere. See also Line 85.

L83 – Turner et al. (Proc. Natl. Acad. Sci. USA. 114, 5367–5372) claim huge shifts in OH are

taking place. I think that's nonsense but it's strongly held by Harvard/Caltech...Rigby also calls for OH shifts. This needs a 1-sentence discussion, rather than being swept into the carpet as refs in parentheses)

L135 – inventories – refs needed.

L150-3. Good point.

L153 – China's coal emissions have risen unabated. Here we hit the isotope discussion. Even though he's an author, there is no mention of Schwietzke et al 2016, or Nisbet et al. 2016 etc on fossil vs biogenic. Authors should consider inserting a paragraph discussing the isotopic evidence. It rather looks as if the authors are running away from mentioning the isotopic shift to lighter values as it conflicts with their conclusions.....

The following papers might be relevant:

Nisbet, E. G., et al. (2016), Rising atmospheric methane: 2007–2014 growth and isotopic shift, *Global Biogeochem. Cycles*, 30, doi:10.1002/2016GB005406.

Schwietzke et al. (2016) Upward revision of global fossil fuel methane emissions based on isotope database. *Nature* 538, 88-91.

Schaefer, H., et al., 2016: A 21st century shift from fossil-fuel to biogenic methane emissions indicated by $^{13}\text{CH}_4$. *Science*, 352, 80-84.

Worden, J. R., Bloom, A. A., Pandey, S., Jiang, Z., Worden, H. M., Walker, T. W., ... & Röckmann, T. (2017). Reduced biomass burning emissions reconcile conflicting estimates of the post-2006 atmospheric methane budget. *Nature communications*, 8(1), 2227.

Zazzeri, G., Lowry, D., Fisher, R.E., France, J.L, Lanoisellé, M., Kelly, B.F.J., Necki, J.M., Iverach, C.P., Ginty, E., Zimnoch, M., Jasek, A., and Nisbet, E.G. (2016) Carbon isotopic signature of coal-derived methane emissions to atmosphere: from coalification to alteration. *Atmos. Chem. Phys.* 16, 13669-13680. doi:10.5194/acp-16-13669-2016

L192 – mention the top down vs bottom up conflict (Saunois et al, Kirschke et al).

Replies to the reviewers

We would like to thank the reviewers for their thoughtful feedback and suggestions on the manuscript. We have revised the manuscript according to these suggestions and have included replies to the reviewer comments below. The original reviewer comments are in black and the replies in blue.

Reviewer #1

The topic fits the scope of the journal, and research work presented in this paper may have significant implications. The developed dataset could also be valuable for others working on related topics.

This manuscript however do have a few places that may need to be improved:

a. This manuscript is reasonably well-written. Though there are still room for further writing improvements.

Thank you for the feedback. We hope that the edits detailed throughout this document have improved the manuscript.

b. The introduction, whilst informative, is very long and should be reduced and prioritized.

We have tried to keep the introduction as brief as possible while still including all of the information and context requested by the other two reviewers.

c. Where did you refer 72% in line 17, page 1.

This statistic is from the Energy Information Administration. We have re-arranged the parenthetical citations in this paragraph to make the attribution of this statistic clearer (line 18 of the revised manuscript).

d. There are lots of contents in the method section. To better guide readers, I would recommend rewriting/adding one paragraph at the beginning of section 2, 3 and 4 to provide an overview of the method subsection, and to help readers organizing their thoughts.

We have divided the methods within the main article and the Supplement into more sub-sections to better guide the reader. These shorter sections break up the methods into more digestible, concise topic areas.

e. Please define the units here where you first mention them, as opposed to further down the page.

We have looked for instances of this issue and corrected it in the manuscript where applicable.

f. References should be listed in the order in which they are quoted. Where are the reference 18th and 19th?

We have corrected this issue in the revised manuscript.

g. In supplemental information, Page 1, the last line, please delete “both”. Where are Fig.2b and panel (b)?

We accidentally labelled two panels of Fig. S2 with the letter “a”. We have corrected the panel labels. We have also deleted the word “both.”

h. What are these errors? Are these just proportional differences? Please define

We discuss potential errors in the GOSAT observations in Sect. S1.2. We also discuss the error covariance matrices used in the inverse model in Sect. S3.2. We have added additional description in both sections to clarify these errors, how they are structured, and what they represent (lines 35-50 the Supplement and Sect. S3.2 of the Supplement).

i. Please add a discussion on the limitations of the study, for completeness. In particular the data is very opaque, so a discussion around this would be useful.

The manuscript now includes greater discussion of several sources of uncertainty – uncertainties due to the hydroxyl radical (lines 99-103 of the main manuscript and Sect. S4.4 of the SI), model-data comparisons at in situ observation sites (lines 114-119 of the main manuscript), and greater detail on the latitudinal bias between GEOS-Chem and GOSAT observations (lines 235-241 of the main manuscript and Sect. S1.2 of the SI). The revised manuscript also further discusses the results in the context of recent studies of methane isotope observations and associated uncertainties in the global fossil fuel methane budget (Nisbet et al. 2016, Schaefer et al. 2016, Schwietzke et al. 2016) (Sect. S4.3).

j. In line 110, “Additional lines of evidence also indicate that coal is likely driving the overall trend in China’s emissions (from 3,400 to 4,000 million metric tons).” Actually, the data on coal production should be questionable.

On page 8, lines 188-194 of the original manuscript (lines 196-198 of the revised manuscript), we point out that coal production numbers are uncertain and cite two sources that discuss recent revisions to China’s coal consumption statistics. In the revised manuscript, we have now included two different coal production estimates in Fig. 6 – official Chinese government estimates (National Bureau of Statistics 2017) and production estimates from the US Energy Information Administration (EIA 2017).

k. Many readers may aware that China has been aggressively targeting small scale coal mining operations and have closed many small coal mines. The trend will likely to persist in the next few years. Will this policy impact the author’s results?

We have also read about these changes and had included a discussion of this trend in an earlier draft of the manuscript. However, the effect of this trend on methane emissions seemed unclear and speculative, so we removed this information from the introduction when iterating on the manuscript drafts. Hence, we have opted to leave this information out of the manuscript.

References

Nisbet, E.G. et al. Rising atmospheric methane: 2007–2014 growth and isotopic shift. *Global Biogeochemical Cycles* 30, 1356-1370 (2016).

Schaefer, H. et al. A 21st-century shift from fossil-fuel to biogenic methane emissions indicated by ^{13}C . *Science* 352, 80-84 (2016).

Schwietzke, S. et al. Upward revision of global fossil fuel methane emissions based on isotope database. *Nature* 538, 88-91 (2016).

National Bureau of Statistics of China China Statistical Yearbook 2017 Ch. 9 (China Statistics Press, Beijing, China, 2017). <http://www.stats.gov.cn/tjsj/ndsj/2017/indexeh.htm>. Last access: 11 July 2018.

US Energy Information Administration International energy statistics. <https://www.eia.gov/beta/international/data/browser/> (2017). Last access: 19 Mar 2018.

Reviewer #2

General comments

This manuscript presents an analysis of recent, 2010 to 2015, trends in methane emissions from China based on GOSAT satellite retrievals of methane. The authors find a significant increasing trend over the study period, most likely from coal, which is in accordance with “business as usual” scenarios and indicates that policy changes to reduce emissions have had no notable impact on emissions. This study is scientifically sound and generally well presented. However, there are a few minor points that need clarification before publication.

Specific comments

L30: Please state the version of the EDGAR inventory, as the estimate has been revised with each version.

This text refers to EDGAR v4.3, and we have specified the version in the revised manuscript.

L72: The trend is actually in sub-tropical Asia and temperate Asia and in tropical Africa.

We have updated the text in the manuscript accordingly.

L98-99: I suggest including the comparisons with observations in the manuscript, as this is an essential part of the method validation.

We have moved the comparison with in situ observations from the SI to the main manuscript (Fig. 4 and lines 114-119).

L104-109: I find it a bit misleading to say that this study finds a clear trend in the emission from coal since this finding is based on the total trend and the fact that in the EDGAR inventory coal is the largest source sector. So even if the share of the total emissions across all sectors is constant in time, coal would show the largest trend; what the authors find is a trend in the total which is likely owing to coal since this likely is the largest source sector.

We have clarified the approach to source attribution within the revised manuscript (lines 265-278). We do not attribute the trend proportionally to the total overall contributions of different sectors in the EDGAR emissions inventory. Rather, we attribute emissions within each individual model grid box based upon the fraction of emissions that is due to each sector within that grid box in the EDGAR emissions inventory. The results in Fig. 4 of the original manuscript (now Fig. 5 of the revised manuscript) displays the summed total of all grid boxes. Note that the trends in this figure are not proportional to the total overall contributions of different sectors in EDGAR. Rather, coal emissions from China increase through time while emissions from other sectors are relatively flat.

In addition, we also examined trends in agricultural production and coal production across China; coal mining increased during the study time period while ruminant populations and rice production declined or remained flat. It is difficult to envision a case in which ruminants or rice production would have been driving the emissions trend, as production numbers for those sectors did not increase during the study period.

Therefore, both the spatial pattern of the emissions trend and economic activity data provide relatively convincing evidence that coal is the most likely sector driving the trend.

L141-143 and Fig. 5: Interestingly the UNFCCC emission factor estimate shows an increase between 2005 and 2012, which is in contrast with the EPA and EDGAR inventories and with what is expected based on policy changes. Can the authors comment on why this may be? We have updated Fig. 5 (now Fig. 6) with more representative coal production numbers. Specifically, the Chinese government does not publish official coal emissions factors in an English language publication. Rather, the government describes its overall approach to developing emissions factors in their second national communication to the UNFCCC (<https://unfccc.int/resource/docs/natc/chnnc2e.pdf>). To calculate the emissions factors in Fig. 5 (Fig. 6 of the revised manuscript), we divided China's coal methane emissions reported to UNFCCC by total coal production. We had initially used coal production estimates from the US Energy Information Administration. In the revised manuscript, we have instead divided by China's official coal production numbers, which are slightly different from EIA numbers. In the revised figure, the emissions factors associated with the UNFCCC numbers are relatively flat. We point out this distinction between coal production estimates in lines 202-207 of the revised manuscript and in Fig. 6 of the revised manuscript.

We suspect that the Chinese government used the same emissions factors for its 2005 and 2012 methane emissions reporting to the UNFCCC. The Chinese National Development and Reform Commission describes its general approach to calculating UNFCCC-compliant emissions factors in its second national communication to the UNFCCC (National Development and Reform Commission, 2012). It estimated province-specific emissions factors using available coal mine data collected across the country. This document includes emissions estimates for 2005, and China later reported its emissions for 2012 to UNFCCC. We suspect that China used the same provincial emissions factors for this 2012 update. Any changes in the emissions factor in Fig. 6 of the revised manuscript more likely reflect changes in coal production from provinces with less gassy coal to more gassy coal. We have added a short discussion of this point to lines 202-207 in the revised manuscript.

Fig.1: The absence of, and in some areas even decreasing, trend in the US is important as it contrasts with several recent studies. Currently this is only discussed in the SI. Even though the focus of this study is Asia, the authors should consider moving this to the manuscript as it's an important result.

We have moved additional discussion on this figure from the SI to the main manuscript (lines 80-87). Figure 1 suggests that the trend in XCH₄ in some parts of the US is less than the global background trend. It does not necessarily indicate a decreasing trend in emissions or XCH₄. With that said, we feel that this topic is beyond the scope of the current manuscript and are hesitant to wade into the existing debate. For example, existing studies have come to very different conclusions on whether there is a trend in emissions from the US (e.g., Turner et al. 2016, Bruhwiler et al. 2017). The purpose of this figure is to show a trend in XCH₄ across East Asia that is anomalous relative to other regions of the globe. It is not intended to provide an exhaustive analysis of US emissions.

SI, section 2: How were the OH data used in GEOS-Chem obtained, and were these pre-optimized using another tracer?

We use archived OH fields from full chemistry GEOS-Chem runs, the same fields used in Turner et al. (2015) and Wecht et al. (2014). Note that we have also included a short discussion of uncertainties due to OH at the request of reviewer #3 in lines 99 to 103 of the main manuscript and Sect. S4.4 of the SI.

SI, section 1.3: The authors state that GOSAT displays a latitude dependent bias compared to modelled XCH₄ (using bottom-up emission estimates). Which model is this and how can the authors be sure that the model-observation discrepancy is not due to the model, such as latitudinal dependent biases in the emission inventories, in the transport or in the atmospheric loss of CH₄? How do the bias corrections change the comparison to TCCON data?

We have added additional detail on this point in Sect. S1.2 of the SI. Several existing studies have identified a similar latitude-dependent difference between GOSAT and modeled XCH₄ (Fraser et al. 2013, Turner et al. 2015). Both of those studies use University of Leicester proxy retrievals and GEOS-Chem. The present study, by contrast, uses the REMOTEC proxy retrieval. In other words, the model—data difference is not unique to the specific retrieval used in this study.

Turner et al. (2015) examine and diagnose this difference in detail relative to existing in situ and TCCON observations. Global GEOS-Chem simulations show a minimal overall bias (~4ppb) and no latitudinal bias when compared to HIPPO (HIAPER Pole-to-Pole Observations) observations. Turner et al. (2015) further compare global GEOS-Chem outputs at against TCCON and report R² of 0.82 (prior model) and 0.83 (posterior model), and mean biases 6.4ppb and 8.1ppb, respectively. These biases are generally smaller than the latitudinal differences between GOSAT and GEOS-Chem (e.g., Fig. S2 of the present paper).

Turner's analysis, and the fact that the GEOS-Chem – GOSAT bias is large in very remote regions like the South Pole, indicates that this bias is unlikely due to the emissions inventories. By contrast, Turner et al. (2015) speculate that this bias is either due to biases in GEOS-Chem in the stratosphere or due to a latitude-dependent bias in GOSAT. Fraser et al. (2013) speculate that the model—GOSAT differences could be due to cirrus clouds, sensitivity of the satellite to zenith angle, and/or errors in modeled CO₂ used in the proxy retrieval.

SI, section 3.1: lambda, i.e. the optimized scaling factors, are the unknown quantity but mu, the prior scaling factors, must be assigned for the inversions, i.e. the prior must be known.

We have clarified this point in the revised SI on lines 119-126. The prior scaling factor (lambda) is unknown in the inverse modeling setup in this paper. This setup ensures that the prior flux model is unbiased relative to the posterior flux estimate, a common statistical assumption of inverse modeling.

SI, section 6: Figure 8c&d shows a significant decrease in CH₄ mixing ratio between 2012-2014. Is the decrease reflecting global variations? What do the authors think is causing the decrease as it contrasts with the generally increasing CH₄ emissions from China.

There was an error in the script that we used to create the timeseries; we omitted a transpose in the original plotting script. As a result of this coding error, the points on the timeseries plots were in the wrong chronological order. We have corrected the plot in the revised manuscript. There is

no decline in CH₄ mixing ratios at the Taiwan sites in the corrected figure (Fig. 4 in the revised manuscript).

Technical comments

L2: I don't think "plurality" is the right term here but rather "majority"

Emissions from coal mining constitute about a third of China's total anthropogenic methane emissions. Coal is the likely the largest anthropogenic methane source in China but does not quite account for a majority of emissions. We have changed this term to "largest fraction" (line 2).

L4: "went" should be replaced by "came"

We have changed this text accordingly.

L85: "Indian" (not India) emissions

We have changed this text accordingly.

References

Bruhwyler, L. M. et al. U.S. CH₄ emissions from oil and gas production: Have recent large increases been detected? *Journal of Geophysical Research: Atmospheres* 122, 4070–4083 (2017). 2016JD026157.

Fraser, A. et al. Estimating regional methane surface fluxes: the relative importance of surface and GOSAT mole fraction measurements. *Atmospheric Chemistry and Physics* 13, 5697–5713 (2013).

National Development and Reform Commission of the People's Republic of China. Second National Communication on Climate Change of The People's Republic of China (2012). URL <https://unfccc.int/resource/docs/natc/chnnc2e.pdf>. Last access: 20 July 2018.

Turner, A. J. et al. Estimating global and North American methane emissions with high spatial resolution using GOSAT satellite data. *Atmospheric Chemistry and Physics* 15, 7049–7069 (2015).

Turner, Alex J., et al. "A large increase in US methane emissions over the past decade inferred from satellite data and surface observations." *Geophysical Research Letters* 43.5 (2016): 2218-2224.

Wecht, K. J., Jacob, D. J., Frankenberg, C., Jiang, Z. & Blake, D. R. Mapping of North American methane emissions with high spatial resolution by inversion of SCHIAMACHY satellite data. *Journal of Geophysical Research: Atmospheres* 119, 7741–7756 (2014). 2014JD021551.

Reviewer #3

This is a very interesting and potentially controversial paper, that attempts to show that China's efforts to curb methane emissions have not worked.

The problem is very important, with major implications for the UN FCCC Paris Agreement. Thus the paper makes an extremely valuable contribution and strongly deserves to be published.

That said, there are major weaknesses in the paper that need to be fixed or at least duct-taped with weasel words.

1. There is no mention of the isotopic evidence for a global shift to lighter values, even though Schwietzke is an author and Turner et al and Rigby et al are mentioned (and the likelihood of an OH collapse dismissed!). Increased isotopically heavy Chinese coal methane would drive $\delta^{13}\text{C}$ strongly more heavy: it is possible to accommodate this in a global budget where the fossil fuel share is dropping, but the impact on the global isotopic balance needs to be discussed.

We have added text to the manuscript reconciling the results of this study with recent studies on $\delta^{13}\text{C}$ methane isotopes (Sect. S4.3). Atmospheric observations of methane isotopes indicate that total global fossil fuel CH_4 emissions have remained relatively flat, in spite of increasing global CH_4 levels (e.g., Schwietzke et al. 2016). However, Schwietzke et al. (2016) argue that decreasing natural gas CH_4 emissions at global scale have been compensated by increasing coal emissions to produce a flat global trend in total fossil fuel CH_4 emissions (Fig. S10 in Schwietzke et al. 2016). Natural gas operations, they argue, have become more efficient over time, and leak rates have decreased from a global average of 8% to 2% over the past 30 years. By contrast, it is more likely that emissions factors from coal operations have remained unchanged during the same time period, and total coal CH_4 emissions have increased as total natural gas CH_4 emissions have declined. Furthermore, the increase in coal CH_4 emissions from China for 2010-2012 estimated in this study is less than that the global coal emissions increase estimated in Schwietzke et al. (2016) for the same time period. Hence, the emissions estimated here for China are not inconsistent with trends in atmospheric $\delta^{13}\text{C}$ observations.

2. There is no discussion of what is going wrong in China – as the paragraph from line 37-44 makes clear, the Chinese government has made major changes to the policy framework; Why is this policy failing?

We have added additional text to the article discussing possible reasons why China's methane reduction policies have not worked (lines 151-180). Governmental agencies like the US EPA Coalbed Methane Outreach Program and organizations like the International Energy Agency have produced documents detailing the potential challenges that China faces in reaching its coal methane targets. One or more of these challenges has presumably hampered China's efforts to curb coal methane emissions to date.

Conclusion – yes, this is an important paper and definitely should be published. But it needs to be armor-plated against the charge that it jumps to conclusions. Thus

"Accept subject to moderate-scale revision."

Thank you again for the helpful feedback and suggestions on the manuscript. We completely agree that it will be important to armor-plate the article against potential criticisms.

Specific comments

Abstract Line 2 – “plurality”: maybe this word will be tough for a non-Anglophone to understand – US/UK election meaning is the largest share but not a majority, but many readers, especially from PR democracies will assume it means ‘majority’. Also say anthropogenic. I know this is the quasi abstract but this statement needs to cite a reference either here or in next para – Peng et al? Atmos. Chem. Phys., 16, 14545-14562. The Peng paper should be cited somewhere: they get 45Tg anthropogenic CH₄ in 2010.

Reviewer #2 also recommended changing the word “plurality”, and we have changed the abstract accordingly (line 2). We have also added the term “anthropogenic” in the abstract; we completely agree that it is a good idea to be very clear in that regard.

We double-checked the editorial guidelines for *Nature Communications*, and they do not allow citations in the abstract. Were it not for this policy, we would include a reference for this sentence of the abstract.

L17 – 72% in which year? Needs a ref.

In 2015. We have updated this sentence and re-arranged the parenthetical citations to make the attribution clearer.

L21 – world’s largest emitter. Well, maybe but how about Brazil and its wetlands? In general this paper makes no distinction between natural and anthropogenic methane. That’s a mistake – the paper needs to be very clear exactly when it is talking about anthropogenic emissions only, and when it means total emissions. Also, the ref here is EDGAR, but there is a huge discrepancy between EDGAR and top-down estimates.

This a great point. We have made sure to be very clear when we are referring to anthropogenic emissions versus total emissions (I.e., anthropogenic plus natural) throughout the revised manuscript. We have also added a caveat to the manuscript (“according to some estimates”) to indicate the uncertainty in greenhouse gas inventories like EDGAR (line 24).

L24-25 majority in coal mining? Is that proven? – not in power station pulverising? REF needed. We have added two references to this section to bolster that statement (line 28).

L25 “and so have coal-related emissions”. – no documentation for this bald statement. It is not necessarily true – the task of the paper is to prove that China’s coal industry reforms have not made it better at capturing methane leaks. This line is tantamount to saying the accused is guilty before hearing the evidence.

Good point. We have removed the statement in quoted by the reviewer from the manuscript.

L35 – note the Thompson study uses isotopes. That’s the big omission in this paper.

We have added a section to the supplement that discusses isotopes (Sect. S4.3).

L63 and L70-75 – GOSAT - worth mentioning Parker, R.J., et al. (2018) Evaluating year-to-year anomalies in tropical wetland methane emissions using satellite CH₄ observations. Remote Sensing of the Environment, 211, 261-273.

We have added this reference to in the paragraph in line 79 of the revised manuscript.

Fig. 1 – probably should have a link to Supplementary online material about the methodology of GOSAT retrievals, or at least a ‘go-to’ reference.

We have included a reference and link to the REMOTEC retrieval methodology manual within the Methods section of the main manuscript (lines 227-228).

Fig. 2 – difference with the Ganesan study is very interesting. Needs some discussion in text somewhere. See also Line 85.

The total methane emissions (anthropogenic and natural) estimated by this study for India are in the mid-range of several inverse modeling inter-comparison projects, while the emissions estimated in Ganesan et al. (2016) are well below any existing top-down estimate (Kirschke et al. 2013, Sounois et al. 2016). For example, we estimate total anthropogenic and natural Indian methane emissions at $\sim 36 \text{ Tg yr}^{-1}$ (2010-2015 mean) while the 9 inverse models in Kirschke et al. (2013) estimate a mean budget of 33 Tg yr^{-1} for 2000-2009 (multi-model mean). The smallest of the nine inverse modeling estimates in Kirschke et al. (2013) is larger than the emissions estimated by Ganesan et al. (2017) (22.0 Tg yr^{-1}), suggesting that the latter study is an outlier.

Similarly, Sounois et al. (2016) report the results of 14 different inverse modeling estimates that use different transport models and different methane datasets. The mean of the 14 estimates is 39 Tg yr^{-1} (2003-2012 mean), and the minimum of the estimates is 37 Tg yr^{-1} . These numbers are similar to the present study but well above the 22.0 Tg yr^{-1} budget estimated by Ganesan et al. (2017).

We suspect that the difference between Ganesan et al. (2016) and existing inverse modeling studies could be due to that study’s methane boundary condition. The authors of that study use model simulations from the MOZART model to estimate methane mixing ratios in air entering the regional modeling domain. The authors then optimize methane emissions to match atmospheric observations minus this boundary condition. The study authors use the EDGAR v4.2 and GFED estimates to drive anthropogenic and biomass burning emissions the MOZART simulations, and the authors do not optimize the boundary condition or evaluate it against atmospheric observations. Existing studies indicate that methane emissions inventories like EDGAR v4.2 are too high across tropical Africa and China (e.g., Turner et al. 2015). The boundary condition in Ganesan et al. (2017) could therefore overestimate methane mixing ratios. If the model boundary condition is too high, it would bias the estimated emissions low. In the present study, by contrast, we optimize emissions for all regions of the globe, albeit at coarse resolution, to ensure there is no bias in the flux estimate due to a fixed boundary condition.

In the revised paper, we have added a sentence pointing out that Ganesan et al. (2016) is an outlier compared to existing inverse modeling estimates in Sounois et al. (2016) and Kirschke et al. (2013) (lines 120-125). We also point out that our estimate is in the mid-range of these existing inter-comparison studies. We are hesitant to add any further discussion of Ganesan et al. (2016) to the revised manuscript. We agree that it is important to establish that the results of the present manuscript are reasonable, in spite of the lower numbers in Ganesan et al. (2016). However, we are also not explicitly trying to critique that study, and we hesitate to speculate within the manuscript on why the emissions in Ganesan et al. (2016) are so low.

L83 – Turner et al. (Proc. Natl. Acad. Sci. USA. 114, 5367–5372) claim huge shifts in OH are taking place. I think that’s nonsense but it’s strongly held by Harvard/Caltech... Rigby also calls

for OH shifts. This needs a 1-sentence discussion, rather than being swept into the carpet as refs in parentheses)

The reviewer makes a great suggestion here. We have added additional explanation to the text (lines 99-103 of the main article and Sect. S4.4) of the SI). Possible changes in OH would yield a small trend in XCH₄ relative to those observed across China from the GOSAT satellite. We discuss this point in detail in Sect. S4.4 of the SI. Furthermore, let us suppose that changes in OH were responsible for the upward trend in GOSAT observations across China. If this scenario were true, then emissions would concomitantly need to be decreasing across Mongolia, Central Asia, and parts of Russia, because GOSAT observations across those regions have been increasing less quickly than in China and less quickly than the NOAA marine background. This scenario seems unlikely. Rather, it appears far more likely that the anomalous upward trend in GOSAT XCH₄ across China is due to changes in emissions, not regional perturbations to OH.

L135 – inventories – refs needed.

We have added references to this line. We discuss this topic in the introduction and added references from that discussion.

L150-3. Good point.

Thanks!

L153 – China’s coal emissions have risen unabated. Here we hit the isotope discussion. Even though he’s an author, there is no mention of Schwietzke et al 2016, or Nisbet et al. 2016 etc on fossil vs biogenic. Authors should consider inserting a paragraph discussing the isotopic evidence. It rather looks as if the authors are running away from mentioning the isotopic shift to lighter values as it conflicts with their conclusions.....

We have added a section to the Supplement on isotopes (Sect. S4.3), and we have added citations to both Schwietzke et al. (2016) and Nisbet et al. (2016). We appreciate this suggestion and agree that the overall conclusions are now more resilient with a discussion of isotopes.

The following papers might be relevant:

Nisbet, E. G., et al. (2016), Rising atmospheric methane: 2007–2014 growth and isotopic shift, *Global Biogeochem. Cycles*, 30, doi:10.1002/2016GB005406.

Schwietzke et al. (2016) Upward revision of global fossil fuel methane emissions based on isotope database. *Nature* 538, 88-91.

Schaefer, H., et al., 2016: A 21st century shift from fossil-fuel to biogenic methane emissions indicated by ¹³CH₄. *Science*, 352, 80-84.

Worden, J. R., Bloom, A. A., Pandey, S., Jiang, Z., Worden, H. M., Walker, T. W., ... & Röckmann, T. (2017). Reduced biomass burning emissions reconcile conflicting estimates of the post-2006 atmospheric methane budget. *Nature communications*, 8(1), 2227.

Zazzeri, G., Lowry, D., Fisher, R.E., France, J.L, Lanoisellé, M., Kelly, B.F.J., Necki, J.M., Iverach, C.P., Ginty, E., Zimnoch, M., Jasek, A., and Nisbet, E.G. (2016) Carbon isotopic

signature of coal-derived methane emissions to atmosphere: from coalification to alteration. *Atmos. Chem. Phys.* 16, 13669-13680. doi:10.5194/acp-16-13669-2016

We have added all of these references in the discussion on methane isotopes.

L192 – mention the top down vs bottom up conflict (Saunois et al, Kirschke et al).
We have added a sentence to this section to highlight this conflict.

References

Ganesan, A. L. et al. Atmospheric observations show accurate reporting and little growth in India's methane emissions. *Nature Communications* 8, 836 (2017).

Kirschke, S. et al. Three decades of global methane sources and sinks. *Nature Geoscience* 6, 813 EP – (2013).

Nisbet, E.G. et al. Rising atmospheric methane: 2007–2014 growth and isotopic shift. *Global Biogeochemical Cycles* 30, 1356-1370 (2016).

Peng, S. et al. Inventory of anthropogenic methane emissions in mainland China from 1980 to 2010. *Atmospheric Chemistry and Physics* 16, 14545–14562 (2014).

Saunois, M. et al. The global methane budget 2000–2012. *Earth System Science Data* 8, 697–751 (2016).

Schwietzke, S. et al. Upward revision of global fossil fuel methane emissions based on isotope database. *Nature* 538, 88-91 (2016).

Turner, Alexander J., et al. "Ambiguity in the causes for decadal trends in atmospheric methane and hydroxyl." *Proceedings of the National Academy of Sciences* 114, 5367-5372 (2017).

REVIEWERS' COMMENTS:

Reviewer #1 (This reviewer only left remarks to the Editor)

Reviewer #2 (Remarks to the Author):

This study is an important constraint on CH₄ emissions from China, which represent a significant fraction of the global anthropogenic CH₄ emission, and points out that the measures put in place to mitigate the emissions, especially from coal, are having no measurable impact. The revised paper has addressed most of the concerns raised in the first review and is overall scientifically sound and well presented. However, I have a few remaining minor comments.

(Note page numbers refer to the manuscript with tracked changes)

Fig. 4. The inversion only assimilated GOSAT data. So the observations shown in Fig. 4 are independent (i.e., not assimilated). This should be mentioned in the figure caption or main text. Also the caption should mention where the data are from (name the database or network).

Fig. 5. Is it possible to put uncertainty bars on these estimates?

L228: I don't see how Fig. 5 relates to the statement "estimated CH₄ emissions increases are highest in regions where the EDGAR inventory indicates a predominance of coal mining emissions relative to other source types"

L206: Please state how large the bias correction was in East Asia. Although this is given in the SI it should be mentioned in the main text as well. Did the authors also check if the bias is constant in time? Given the trend analyses are the focus of this paper, it would be important to confirm this.

L107 of SI: Should the uncertainty for China be ± 0.358 Tg/y (rather than 3.58 Tg/y)?

L143-148 of SI: The authors state that their inverse problem set-up means that the prior model is unbiased relative to the GOSAT observations. If I understand correctly, the method optimizes a vector of scalars of the prior emissions, μ , as well as the mean value of μ ? If so, I think it would be clearer if a different symbol was used to indicate the variable "mean of μ ". Also, how this is done is not clear from Eq. S2, or is this done iteratively?

L275-277 of SI: This statement is not correct. All the d¹³C isotope observations can tell is that either: 1) the d¹³C value of the total source is decreasing (which does not necessarily mean that fossil fuel emissions are not increasing, see e.g. Worden et al. 2017) or 2) that the atmospheric sink of CH₄ has decreased, or 3) a combination of these. I can perfectly agree with the later statement "the emissions estimated here for China are not inconsistent with the trends in atmospheric d¹³C observations" but the opening statement of this section should be amended.

Reviewer #3 (Remarks to the Author):

The authors have responded well and in detail to the points raised. The paper is important and should now go forwards to publication.

Reply to the reviewers

We would like to thank the reviewers for their thoughtful suggestions and ideas throughout the review process. The reviewers have helped us improve both the analysis and the manuscript. Below, we have replied to additional comments and suggestions from reviewer #2. The reviewer's comments are highlighted in blue and our responses are written below each comment.

This study is an important constraint on CH₄ emissions from China, which represent a significant fraction of the global anthropogenic CH₄ emission, and points out that the measures put in place to mitigate the emissions, especially from coal, are having no measurable impact. The revised paper has addressed most of the concerns raised in the first review and is overall scientifically sound and well presented. However, I have a few remaining minor comments.

(Note page numbers refer to the manuscript with tracked changes)

Fig. 4. The inversion only assimilated GOSAT data. So the observations shown in Fig. 4 are independent (i.e., not assimilated). This should be mentioned in the figure caption or main text. Also the caption should mention where the data are from (name the database or network).

This statement is correct, and we have added text to the manuscript accordingly (in the section *Trends in emissions from China and Asia*). The data are from the NOAA Global Greenhouse Gas Reference Network (<https://www.esrl.noaa.gov/gmd/ccgg/>), and we have added that information to the figure caption.

Fig. 5. Is it possible to put uncertainty bars on these estimates?

There are no uncertainty bounds included in the EDGAR emissions inventory, so it would be difficult to construct realistic uncertainty bounds for this figure. The sector attribution in Fig. 5 is based upon the relative portion of emissions attributed to each sector at each location in the EDGAR emissions inventory. The EDGAR inventory does not list any uncertainties for this sector attribution, precluding us from estimating uncertainties in Fig. 5. With that said, we do estimate uncertainty bounds on the total emissions estimated by the inverse model, shown in Figs. 2 and S6.

We have added text to the Fig. 5 caption explaining this point and referring the reader to the uncertainty estimates in Figs. 2 and S6.

L228: I don't see how Fig. 5 relates to the statement "estimated CH₄ emissions increases are highest in regions where the EDGAR inventory indicates a predominance of coal mining emissions relative to other source types"

We have revised this statement in the text to clarify. The attribution in Fig. 5 is based upon the spatial patterns in EDGAR emissions inventory (described in *Methods*). We estimate an increase in emissions from the coal sector in Fig. 5 because EDGAR assigns large coal emissions relative to other source sectors in regions where we find that emissions are increasing. Figure 5, however, does not display these emissions patterns, only the estimated emissions totals. Figs. S4 and S5 more clearly show these patterns. We have moved the reference to Fig. 5 to the first sentence of the paragraph and have referenced Figs. S4 and S5 in the second sentence of that paragraph.

L206: Please state how large the bias correction was in East Asia. Although this is given in the SI it

should be mentioned in the main text as well. Did the authors also check if the bias is constant in time? Given the trend analyses are the focus of this paper, it would be important to confirm this.

The bias correction ranges from -10 to -15 ppb across East Asia. We have added this information to the section *GOSAT observations* in *Methods*. In addition, the model-data residuals at 20 degrees latitude are consistently around 9ppb higher than 60 degrees latitude for each year of the model simulations. This difference between tropical and temperate regions does not vary by more than 3 ppb between 2010 and 2015.

L107 of SI: Should the uncertainty for China be ± 0.358 Tg/y (rather than 3.58 Tg/y)?

The uncertainty should be 3.58 Tg/y as printed in the manuscript. The emissions estimate between 2010-2012 has much more variability around the trend than the estimate for 2013-2015 (Fig. 2). As a result, the trend line fitted for years 2010-2012 has a much larger uncertainty than the line fitted for 2013-2015.

L143-148 of SI: The authors state that their inverse problem set-up means that the prior model is unbiased relative to the GOSAT observations. If I understand correctly, the method optimizes a vector of scalars of the prior emissions, μ , as well as the mean value of μ ? If so, I think it would be clearer if a different symbol was used to indicate the variable “mean of μ ”. Also, how this is done is not clear from Eq. S2, or is this done iteratively?

We have revised the text to make the variable definitions clearer. The variable μ is the prior estimate of the scaling factors; it is a single value for each 6-month time period of the inverse model. Many existing inverse modeling studies “hard code” this value at 1. Instead, we estimate μ as part of the inverse model to ensure the prior is unbiased. By contrast, the vector λ are the scalars of the emissions. These posterior scaling factors vary from one model grid box to another and are also estimated as part of the inverse model.

In most cases, one can solve an inverse model of this kind using a system of linear equations. The equations are the same as those described in geostatistical inverse modeling studies (e.g., Kitanidis and Vomvoris 1983, Michalak et al. 2004), except that we estimate a set of scaling factors instead of estimating the emissions directly. Note that we do not use the term “geostatistical inverse modeling” in this study because we estimate scaling factors instead of estimating emissions, as all geostatistical inverse modeling studies have done in the past; in the present study, it was much easier to estimate scaling factors than estimating the emissions directly given the atmospheric modeling setup.

In this particular application, we solve the problem iteratively in order to ensure the estimated scaling factors are non-negative (e.g., Miller et al. 2014). We do not describe the minimization algorithm in depth within the SI because a detailed description would require several pages, and this topic is already discussed in detail in Miller et al. (2014). We felt that such a description would distract from the central components of the atmospheric modeling and inverse modeling setup.

L275-277 of SI: This statement is not correct. All the d13C isotope observations can tell is that either: 1) the d13C value of the total source is decreasing (which does not necessarily mean that fossil fuel emissions are not increasing, see e.g. Worden et al. 2017) or 2) that the atmospheric sink of CH4 has decreased, or 3) a combination of these. I can perfectly agree with the later statement “the emissions estimated here for China are not inconsistent with the trends in atmospheric d13C observations” but the opening statement of this section should be amended.

We have edited the first half of this paragraph. The edited paragraph makes it clear that there have been multiple interpretations of isotopic observations in the literature. We also cite Worden et al. (2017) in the edited paragraph.

References

Kitanidis, Peter K., and Efstratios G. Voulgaris. "A geostatistical approach to the inverse problem in groundwater modeling (steady state) and one- dimensional simulations." *Water resources research* 19.3 (1983): 677-690.

Michalak, Anna M., Lori Bruhwiler, and Pieter P. Tans. "A geostatistical approach to surface flux estimation of atmospheric trace gases." *Journal of Geophysical Research: Atmospheres* 109.D14 (2004).

Miller, S. M., A. M. Michalak, and P. J. Levi. "Atmospheric inverse modeling with known physical bounds: An example from trace gas emissions." *Geoscientific Model Development* 7.1 (2014): 303-315.

Worden, John R., et al. "Reduced biomass burning emissions reconcile conflicting estimates of the post-2006 atmospheric methane budget." *Nature communications* 8.1 (2017): 2227.